# The Effects of Downhill Running and Maturation on Histological and Morphological Properties of Tendon and Enthesis in Mice

**DOI:** 10.3390/biology12030456

**Published:** 2023-03-16

**Authors:** Kaichi Ozone, Yuki Minegishi, Yuichiro Oka, Michiaki Sato, Naohiko Kanemura

**Affiliations:** 1Department of Health and Social Services, Health and Social Services, Graduate School of Saitama Prefectural University, Koshigaya 343-8540, Japan; 2492003c@spu.ac.jp (K.O.);; 2Department of Rehabilitation, University of Tsukuba Hospital, Tsukuba 305-8576, Japan; 3Japan Society for the Promotion of Science, Tokyo 102-0083, Japan; 4Department of Physical Therapy, Health and Social Services, Saitama Prefectural University, Koshigaya 343-8540, Japan

**Keywords:** enthesis, tendon, downhill running, overuse, eccentric contraction

## Abstract

**Simple Summary:**

Tendon and enthesis disorders account for more than half of all sports injuries in adolescents and adults, primarily resulting from overuse. In mice, downhill running is a common intervention to induce overuse. In this study, we examined how changes in the tendons and entheses are induced by different exercise conditions, such as downhill and flat-land running, in mice of different ages, including adolescent and adult mice. We found that downhill running induced hypertrophy of the muscle–tendon complex and enthesis, as well as a significant increase in inflammatory factors in the enthesis, regardless of the age of the mice. In addition, inflammatory factors are expressed in tendons and enthesis due to downhill running at all age group, but the genetic level suggests that the expression change of inflammatory factors were more pronounced in the tendons of adult mice than in the enthesis of adolescent mice. These results will help elucidate the pathogenesis of enthesopathy and tendinopathy, which have different pathophysiologies despite having the same pathogenetic factors.

**Abstract:**

To date, it remains unclear how overuse affects the tendons and entheses at different stages of maturation. Therefore, we evaluated histological and morphological changes in the tendons and entheses in adolescent (4-week-old) and adult mice (8-week-old) by performing flat-land and downhill running exercises. The mice were divided into the Sedentary, High Flat (flat-land high-speed running; concentric-contraction exercise), Low Down (downhill low-speed running; eccentric-contraction exercise), and High Down (downhill high-speed running; eccentric-contraction exercise) groups. Histological changes and inflammatory factor expressions were compared in the entheses and tendons after 4 weeks of exercise. Downhill, but not flat-land high-speed running, induced muscle–tendon complex hypertrophy in both adolescent and adult mice. Histological enthesis changes were induced in both groups during downhill running but were less pronounced in adult mice. Conversely, no significant cell aggregation or fiber orientation changes were observed in the tendon, but increased inflammatory factors were observed in both groups, with significantly higher expression in the tendons of adult mice. Downhill running induced histological and morphological enthesis changes and inflammatory factor increase in the tendons, regardless of running speed variations. These results may help elucidate the pathogenesis of enthesopathy and tendinopathy, which have different pathophysiologies despite having the same pathogenetic factors.

## 1. Introduction

The forces generated by muscles are transmitted to the bones via tendons and their attachments, the entheses, to induce joint movement. Tendon and enthesis disorders, including Osgood–Schlatter disease (OSD), Jumper’s knee, and Achilles tendinopathy, account for approximately 30–50% of all sports disorders and can trigger limitation of activity among athletes [1]. For instance, OSD, a type of enthesopathy, has a relatively adolescent age (8–15 years) of onset [2,3] and is characterized by bone avulsion in the enthesis region, with inflammation and degeneration in the same site and surrounding tissues [4]. In contrast, Jumper’s knee, a type of tendinopathy, has a relatively adolescent to adult age (14–40 years) of onset [5], with inflammatory findings in the tendon and surrounding tissues, angiogenesis, and degeneration of the tendon substance [6]. Interestingly, overuse has been commonly recognized as the root cause of both disorders [7,8]. Moreover, it is likely that the maturity of the individual influences the prevalence and predominant site of a disorder. However, in actual sports practice, it is not yet possible to identify the factors that cause overuse or establish appropriate strategies to prevent its occurrence.

In basic research using rodents, downhill running has been employed to induce overuse. Decline running at a given speed increases the energy absorption at the load-bearing joints [9] and the demand for energy dissipation in the associated muscle–tendon units [10]. Therefore, downhill running is often used as a model to induce eccentric overload for skeletal muscles. Previous studies have suggested that sports-related pathological changes similar to enthesopathy are more strongly influenced by downhill running, as suggested by basic experiments in adolescent mice. Downhill running induces inflammatory factor increase in the entheses and tendons, and significantly increases the cartilage catabolic factor (matrix metalloproteinase-13), cartilage synthesis-related factor (SRY-box9), and calcification-related factors (collagen type X, alkaline phosphatase, and runt-related transcription factor) in the adolescent mice enthesis [11,12]. Similarly, there are some reports stating that tendon disorders are induced by downhill running, which increases inflammatory factors in the tendon tissue and decreases the mechanical strength of tendons [13,14]. The in vivo experimental model using downhill running is an appropriate exercise model to examine the pathogenesis of enthesopathy and tendinopathy. However, the effect of downhill running speed and skeletal maturity on overuse-related histological and morphological changes of tendon and enthesis remains unclear. As maturation clearly influences the composition and mechanical properties of both tendon and enthesis, it is important to understand the effects of mechanical loading as a function of maturation.

Based on the above, we aimed to examine whether any histological and morphological changes occur in the tendon and enthesis complexes between flat-land and downhill running at different speeds. Furthermore, we aimed to prove whether the changes in exercise conditions and maturity affect the changes in the lesion sites in young and adult mice to elucidate the pathogenesis of enthesopathy and tendinopathy in the future.

## 2. Materials and Methods

### 2.1. Target Animals and Exercise Protocol

Forty (male) mice aged between 3 and 7 weeks were obtained (Japan SLC Incorporated, Shizuoka, Japan). After a 1-week period of environmental and exercise adaptation, adolescent (4-week-old) and adult (8-week-old) mice underwent a 4-week exercise intervention.

All mice were housed in pairs in plastic cages maintained at a temperature of 23 ± 1 °C with a 12-h light/dark cycle. During the exercise intervention, a small treadmill was used for smaller animals. Furthermore, 4- and 8-week-old mice were divided into four intervention groups: Sedentary (control), High Flat (high-speed flat-land running; concentric contraction (CC) exercise), Low Down (low-speed downhill running; EC exercise), and High Down (high-speed downhill running; EC exercise) groups. According to a previous study, the inclination angle of the small treadmill was set to a downhill setting to mimic the EC of the target muscle during running [15] (Figure 1A). The upper limit of the intervention speed was set as 35 m/min at the end of the intervention, because many 8- and 12-week-old mice were dropped out at speeds faster than 35 m/min during the 1-h dropout test. This intervention speed was faster than that generally used for high-intensity exercise (25 m/min) [16]. The intervention start-speed was established at 15 m/min as a moderate speed because many 4-week-old mice were dropped out at 20 m/min during the 1-h dropout test. Therefore, the speeds at the start and end of the intervention were set at 15 and 35 m/min, respectively, and the exercise conditions were set to gradually increase with growth. Exercise intervention consisted of 1 h of running per day, 5 days per week for 4 weeks, and then collecting target tissue from the left and right shoulder joints (Figure 1B).

The exercise intervention did not cause prominent weight loss, injury, or eating disorders in the mice, and they remained healthy during implementation. When running downhill at the same speed as running on flat land, the load on the forelimbs is clearly higher in the downhill running. Therefore, a Low Down group was set up to reduce the mechanical load on the forelimbs during downhill running, and the exercise intensity was arbitrarily set so that the speed of the High Flat group and High Down group was 1.5 times that of the Low Down group (Figure 1C).

### 2.2. Sample Tissue Collection

At the end of the 4-week exercise intervention period, cervical dislocation was performed by skilled personnel on 8- and 12-week-old mice under anesthesia with isoflurane aspiration. In the left shoulder joint, the supraspinatus (SSP) muscle and humerus–SSP tendon complex were collected for histological, bone morphological, and molecular analysis. The right shoulder joint was collected for histological enthesis and tendon evaluations.

### 2.3. Histological Evaluations of the Supraspinatus Muscles, Tendons, and Entheses

The SSP muscles of the 8- and 12-week-old mice were submerged in a mixed acetone-isopentane solution at −100 °C for quick freezing. Frozen sections were then prepared at 10-µm intervals using a Leica CM 3050 S cryostat (Leica Microsystems AG, Wetzlar, Germany). Hematoxylin-eosin (HE) staining was performed on frozen sections of the SSP muscle. To compare the mean cross-sectional area (CSA) between the groups, 150 muscle fibers were counted using the Hybrid Cell Count software (BZ-H3C; KYENCE, Osaka, Japan), and the mean CSA per fiber was calculated.

The collected right shoulder joints from the 8- and 12-week-old mice were treated with 4% paraformaldehyde solution for 24 h of fixation and treated with 10% ethylenediaminetetraacetic acid solution for 2 weeks of demineralization. Thereafter, these samples were embedded in paraffin and cut in continuous 5-µm sections. After deparaffinization, HE staining and 0.05% toluidine blue (TB) staining were performed. TB-stained sections were used to determine the fibrocartilage (FC) areas. The FC and calcified fibrocartilage (CFC) areas were determined based on the characteristic cell shape in each region, referring to previous studies [17]. The Fiji image analysis software (US National Institute of Health, Bethesda, MD, USA) [18] was used to calculate these areas.

### 2.4. Immunohistochemical Evaluations of the Supraspinatus Tendon and Enthesis

Deparaffinization sections were washed thrice for 5 min each with phosphate buffered saline (PBS) (pH 7.4). For antigen activation, Proteinase K (Worthington Biochemical Co., Lakewood, NJ, USA)/distilled water (0.2 mg/mL) was added dropwise to the sections, which were then incubated for 15 min. After rewashing with PBS, endogenous peroxidase activity was blocked by BLOXALL blocking solution (Vector Laboratories, Newark, CA, USA) for 10 min. Blocking used 0.05% normal goat serum/PBS solution for 1 h. After rewashing again with PBS, the following primary alternatives were reacted at 4 °C overnight: antitumor necrosis factor-alpha (TNF-α), rabbit polyclonal antibody (1:200 dilution, AF7014; Affinity Biosciences, Solon, OH, USA), and anti-interleukin-6 (IL-6) rabbit polyclonal antibody (1:100 dilution, ab6672; Abcam, Cambridge, MA, USA). Subsequently, the streptavidin-biotin-peroxidase complex technique was performed using an ABC kit (Vector Laboratories). After rewashing with PBS, the sections were colored brown using a 3,3-diaminobenzidine solution (NICHIREI Biosciences, Tokyo, Japan). Nuclei were counterstained with Mayer’s hematoxylin. Immunohistochemically (IHC) stained images were calculated as the ratio of positive cells per unit area using the Fiji image analysis software [18].

### 2.5. Bone Morphological and Quality Evaluations of the Humerus Bone

The humerus-SSP tendon complexes of the 8- and 12-week-old groups were infiltrated in RNAlater^TM^ Stabilization Solution (Invitrogen, Waltham, MA, USA) for 48 h after collection and stored in a freezer at −20 °C. Thereafter, the humerus–SSP tendon complexes were measured using microcomputed tomography (microCT) Sky Scan 1272 (Bruker Corp., Billerica, MA, USA). The measurement setup was as follows: X-ray = 60 kVp/165 µA, detector resolution = 1632 × 1092, rotation angle pitch = 0.1 deg/s, voxel size = 4 µm, and filter setting = 0.25 mm aluminum. The FC and subchondral bone (SB) regions underlying the enthesis CFC area were calculated based on our previous experiment [12]. In addition, the CSA of the tendon was measured at the thinnest region of the tendon mid-substance.

### 2.6. Molecular Biological Evaluations of the Supraspinatus Tendon and Enthesis

Target tissues were immediately harvested from the humerus–SSP tendon complexes after micro-CT imaging was performed, and real-time reverse transcription polymerase chain reaction was used to measure the mRNA relative expression levels of enthesis FC and tendon in the 8- and 12-week-old groups. The StepOne-Plus system (Applied Biosystems, Waltham, MA, USA) were used for analysis. *Tnf-a* (Mm00443258-m1) and *Il-6* (Mm00446190-m1) were used as target primers and *Hypoxanthine phosphoribosyltransferase 1* (*Hprt1*; Mm00446968_m1) as a reference gene (TaqMan Gene Expression Assay prove; Applied Biosystems). Relative gene expression levels were calculated using the 2^−ΔΔCt^ method [19]. The details are described in the Appendix A.

### 2.7. Statistical Analysis

Statistical analysis was performed using JASP 0.16.3 (Intel Corp., Santa Clara, CA, USA) [20,21]. The normality of distribution in each dataset was confirmed using the Shapiro–Wilk test. All data were subjected to a two-way analysis of variance to determine significant effects for each parameter. The main effects and interactions for age in weeks (8- and 12-week-old) and intervention method (Sedentary, High Flat, Low Down, High Down) were determined. Simple main-effect tests were performed on the results for which an interaction was confirmed. The Bonferroni test was used as a post hoc test. All data are presented as means ± standard deviations (SDs), with *p* < 0.05 being statistically significant.

## 3. Results

### 3.1. Downhill Running Exercise Hypertrophies the Supraspinatus Muscle–Tendon Complex

The SSP muscle and tendon CSAs were calculated (n = 5; Figure 2A). The results did not show an increase in muscle CSA in the High Flat group compared to the Sedentary group at 8 and 12 weeks of age (*p* > 0.05). Significant increases in muscle CSA were observed in the Low Down and High Down groups compared to the Sedentary and High Flat groups, regardless of age (*p* < 0.01; Figure 2B). Tendon CSAs were similar to the muscle CSAs, with no increase in tendon CSA in the High Flat group compared to the Sedentary group at 8 and 12 weeks of age (*p* > 0.05). The Low Down and High Down groups demonstrated a significant increase in tendon CSA than the Sedentary and High Flat groups (*p* < 0.01; Figure 2C). In particular, the High Down group had significantly increased tendon CSA compared to all other groups at 12 weeks of age (*p* < 0.01; Figure 2C).

### 3.2. Downhill Running Exercise Induces Histological Changes in the Enthesis Fibrocartilage Areas Depending on Maturity and Slight Changes in the Tissues Surrounding the Tendon

TB staining was performed to examine histological changes in the enthesis FC area (n = 5, Figure 3A). No significant increase in the total FC and CFC area, labeled and calculated by TB staining, was observed between the Sedentary and High Flat groups at 8 and 12 weeks of age (*p* > 0.05; Figure 3B,C). Conversely, in the Low Down group, the total FC area increased significantly at 8 and 12 weeks of age compared to the Sedentary group (*p* < 0.001; Figure 3B), and increased significantly only at 8 weeks compared to the High Flat group (*p* < 0.01; Figure 3B). The High Down group demonstrated a significant increase in total FC area at 8 and 12 weeks of age compared to the Sedentary and High Flat groups (*p* < 0.05; Figure 3B). Regarding the CFC area, only the 8-week-old downhill exercise groups (Low Down and High Down) showed a significant increase in the CFC area compared to the Sedentary and High Flat groups (*p* < 0.05; Figure 3C). Interestingly, for 12-week-old mice, no significant differences in the CFC area were observed between the groups (*p* > 0.05; Figure 3C). Therefore, downhill running exercise affects the uncalcified fibrocartilage (UFC) area and the CFC area up to 8-weeks before maturity, but affects the UFC area only when downhill running is combined with high-speed exercise at 12 weeks of age after maturity. Macroscopic observations of tendon tissue showed no tendinopathy-like pathology, including increased cell number and altered collagen tissue arrangement, between the groups at 8 and 12 weeks of age. However, in the exercise group with downhill running, angiogenesis and synovial membrane thickening were observed in the surrounding tissues, regardless of age (Figure 3A).

### 3.3. High Speed Downhill Running Exercise Induces Changes in Bone Morphology in Adolescents but Has Little Effect in Adults

A bone quality evaluation was performed by microCT imaging (n = 5, Figure 4A). No significant differences in the volume of the humerus head (HH) were observed between the intervention groups in 8- and 12-week-old mice. Conversely, the 12-week-old group showed an increase in volume compared to the 8-week-old groups (*p* < 0.05; Figure 4B). Regarding the FC volume, high-speed downhill running induced an increase at 8-weeks of age (*p* < 0.001), and similar to the HH volume, the FC volume showed a greater volume increase from 8 to 12 weeks of age (*p* < 0.05; Figure 4C). This suggests that both bone tissue growth and FC volume increased with increasing age and that the bone tissue and FC area were affected by the running condition only in adolescence. Regarding the analysis of the subchondral bone region, we observed an increase in the bone volume/tissue volume (BV/TV), trabecular thickness (Tb.Th), and trabecular separation (Tb.Sp; *p* < 0.05; Figure 4D), due to high-speed running condition at 8 weeks of age. Interestingly, these changes were not observed at 12 weeks of age. This suggests that the high-speed downhill running exercise induces morphological changes in the subchondral bone region in adolescent (8-week-old) mice, but has little effect in adult (12-week-old) mice.

### 3.4. Downhill Running Exercise Induces Expression of Inflammation-Related Factors in Enthesis Fibrocartilage and Tendons

IHC staining was performed on the enthesis FC area and tendons to compare the expression of inflammation-related factors. The ratio of positive cells per unit area was calculated (n = 5). TNF-α and IL-6 were labeled as representative inflammatory-related markers and were compared in the FC area (Figure 5A). The percentage of TNF-α-positive cells in the FC area did not significantly increase in the High Flat group compared to the Sedentary group at both 8- and 12-week-old mice (*p* > 0.05), while the High Down group showed a significant increase compared to the Sedentary and High Flat groups (*p* < 0.001; Figure 5B). Furthermore, only at 8 weeks of age, the Low Down group showed a significant increase in the percentage of TNF-α positive cells compared to the Sedentary and High Flat groups (*p* < 0.01; Figure 5B). The percentage of IL-6 positive cells in the FC region did not significantly increase in the High Flat group compared to the Sedentary group at both 8 and 12 weeks of age (*p* > 0.05), while the High Down group showed a significant increase compared to the Sedentary and High Flat groups (*p* < 0.001; Figure 5C). In 8-week-old mice, the Low Down group showed a significant increase in the percentage of IL-6 positive cells compared to the Sedentary and High Flat groups (*p* < 0.01; Figure 5C). Additionally, TNF-α expression in the Sedentary group was significantly reduced as a result of week age (*p* < 0.05; Figure 5B). Although IL-6 expression in the Sedentary group was similar, it was not affected by the age in weeks (Figure 5C).

Following that, we compared the expression of inflammation-related factors in the tendon region (Figure 6A). The percentage of TNF-α-positive cells in the tendon region was significantly increased in the downhill running group compared to the other groups at 8 and 12 weeks of age (*p* < 0.001; Figure 6B). The percentage of IL-6 positive cells in the tendon region significantly increased in the downhill running group compared to the Sedentary and High Flat groups at 8 and 12 weeks of age (*p* < 0.001; Figure 6C), similar to the TNF-α results. At 12 weeks of age, the High Flat group showed a significant increase in both TNF-α and IL-6 expressions compared to the Sedentary group (*p* < 0.05). In addition, both factors significantly decreased only in the Sedentary group due to the effect of age (*p* < 0.05), while IL-6 levels increased significantly in the Low Down and High Down groups (*p* < 0.05; Figure 6B,C).

Confirming the results by site, in the enthesis, high- and low-speed downhill running exercise before maturation significantly increased inflammatory factors compared to flat-land running independent of the running speed, whereas after maturation, inflammatory factors increased by the most stressful exercise (high-speed downhill running). Conversely, in the tendons, both before and after maturation, a downhill running exercise increased inflammatory factors regardless of the running speed.

### 3.5. Inflammatory Factors Were Upregulated in the Fibrocartilages and Tendons in Young Mice and the Tendons in Adult Mice after Running Downhill

To investigate the gene expression changes of inflammatory factors in the enthesis and tendon, we compared the relative gene expressions of *Tnf-α* and *Il-6* in five animals per group (Figure 7A,B). In the FC area, *Tnf-α* expression increased significantly in the Low Down and High Down groups compared to the Sedentary group (*p* < 0.05) and increased significantly in the High Down group compared to the High Flat group (*p* < 0.001) at 8 weeks of age. The results of *Il-6* expression were similar (*p* < 0.01). At 12 weeks of age, *Tnf-α* expression was significantly different between the Sedentary and High Down groups (*p* < 0.05), and the *Il-6* expression was significantly different between the High Down group and Sedentary and High Flat groups (*p* < 0.05). *Tnf-α* and *Il-6* expression levels were not significantly different between the Sedentary and High Flat groups regardless of age (*p* > 0.05; Figure 7A).

In the tendon region, relative *Tnf-α* expression significantly increased in 8-week-old mice in the High Down group compared to the 8-week-old mice in the Sedentary and High Flat groups (*p* < 0.05), and *Il-6* expression was significantly different between the 8-week-old mice in the Sedentary and High Down groups (*p* < 0.05). In contrast, at 12 weeks of age, the Low Down and High Down groups showed a significant increase in *Tnf-α* and *Il-6* expression compared to the Sedentary and High Flat groups (*p* < 0.05). Similar to the results in the FC region, there were no significant differences in *Tnf-α* and *Il-6* between the Sedentary and High Flat groups in the tendon region (*p* > 0.05; Figure 7B), regardless of age.

## 4. Discussion

The results of this study showed that downhill running exercise was associated with inflammatory factor expression in the tendons and entheses of adolescent mice. Furthermore, we confirmed that morphological changes in the FC area may have been induced by downhill running exercise. Conversely, downhill running exercise may also induce inflammatory factor expression in the tendons and entheses of adult mice. The changes were more pronounced in the tendons than in the FC area at the gene level. In adult mice, the CFC area remained unchanged, and morphological changes in the entire FC area were mild. In the tendons, we observed a clear increase in inflammatory factors and minor histological changes in the surrounding tissues, but no obvious tendinopathy-like histological changes in the tendon substance. Therefore, different exercise conditions may induce morphological changes in the enthesis, and the predominance of inflammatory factor expression may differ between the enthesis and tendon depending on the age of the mice.

The histological and morphological changes observed in the present study following downhill running may be the consequence of (a) the eccentric nature of muscle contraction, (b) a greater cumulative mechanical load, and/or (c) a redistribution of loads within the enthesis and tendon. Downhill running has often been employed as a form of locomotion to induce eccentric overload in antigravity muscles in quadrupeds. While eccentric loading induces contraction-type specific acute and chronic effects on muscles, research suggests that this is not the case in tendons, at least for physiological loads. It cannot be ruled out that inflammatory mediators generated in the muscle specific to eccentric loading could have propagated to the tendon tissue and influenced the metabolic load response in the tendon and enthesis [22,23,24]. However, as uphill running (i.e., concentric dominant exercise) can also serve as an overuse model for tendons [25], the main reasons for the observed changes specific to the downhill loading protocols likely relate to the kinematic and kinetic changes associated with downhill running. When quadrupeds are running downhill, the forelimb exerts approximately 84% of the total braking impulse necessary to decelerate the center of mass [26]. Since tendons function as an energy buffer for the muscle, it may be speculated that the mechanical load absorbed by the supraspinatus tendon increased in downhill compared to level running. Moreover, the kinematic changes observed consistently in legged animals running downhill [27] may lead to a redistribution of loads within the tendon and enthesis [28], which may have contributed to the marked histological and morphological changes observed in the downhill groups in the present study, despite the 1.5-fold lower running velocity in the Low Down group compared to the High Flat group.

The present study suggests that downhill running hypertrophies the CSA in the muscle and tendon tissue in mice, regardless of age. Many studies have reported muscle hypertrophy in response to mechanical loading [29,30,31]. However, at a given loading volume, isolated eccentric and concentric loading do not seem to produce significantly different muscle hypertrophy, despite contraction type-specific changes of muscle architecture [32]. Therefore, the more pronounced muscle hypertrophy observed in the downhill running groups was probably the consequence of a greater mechanical load of the shoulder joint and supraspinatus muscle, despite the lower velocity in the Down Low group compared to the High Flat group. In tendon tissues, hypertrophy may be both a physiological adaptive change, as well as a pathological consequence of overuse. Tendon collagen turnover is essentially slow or almost non-existent following maturation in humans and horses [33,34]. Therefore, physiological tendon hypertrophy probably contributes to exercise-induced changes of tendon stiffness in only the long term [35]. An increase in tendon CSA can, however, be the result of overuse and the associated accumulation of fluid and non-collagenous matrix components. For example, Soslowsky et al. reported a hypertrophic response and decreased elastic modulus of rat supraspinatus tendons following downhill running as early as four weeks [13]. Therefore, we interpreted the increase in tendon CSA observed here as a pathological response rather than a physiological adaptation, although no mechanical testing was performed in the present study for confirmation.

The histological enthesis changes in adolescent mice were similar to those observed in previous studies [11,12]. Interestingly, although an increase in the overall area of the FC region occurred even in mature mice, there was no difference between the groups in the CFC region; therefore, it remains unclear whether a marked enthesopathy-like morphological change occurred. The UFC area is located at the site of a significant change in tendon orientation [36], and the UFC, at a site where adaptive responses to mechanical stress, mainly compressive, is likely to occur in the tendons [37,38,39,40]. Therefore, it is possible that downhill running induced greater changes in tendon orientation than level running and caused morphological changes in the enthesis because it generated primarily compressive mechanical stresses in the UFC region [41,42].

The link between inflammation development and morphological changes has been reported in studies related to psoriatic arthritis and spondylarthritis, which are representative diseases of enthesitis. Triggered by mechanical stress or infection, the fibrocartilage induces IL-23 release, which in turn activates ILC3 and γδ T cells [43,44]. Subsequently, γδT cells and ILC3 increase the expression of inflammatory factors, such as TNF-α, IL-17, and IL-22, which induce mesenchymal stem cell activation and osteogenesis-related signaling activation in a mechanical stress-dependent manner, leading to morphological tissue changes [45,46]. Although we were unable to analyze the IL-23 levels in our study, we observed increased TNF-α expression and morphological enthesis changes, and in our previous report, we confirmed activation of osteogenesis-related signaling pathways (BMPs) in the same region [12]. Therefore, it is possible that an inflammatory response triggered by increased mechanical stress specific to downhill running was induced in the enthesis, leading to enthesopathy-like morphological changes.

Finally, the relationship between individual maturation, exercise conditions, the expression of inflammatory factors in the entheses and tendons, and morphological changes was discussed. In adolescent mice, the downhill running group demonstrated an inflammatory response in the FC and tendon regions and morphological changes similar to enthesopathy, regardless of the change in running speed. In contrast, in adult mice, low-speed downhill running did not induce morphological changes in the FC region similar to enthesopathy but caused group differences in the expression of inflammation-related factors, predominantly in tendon tissues. As the only mouse characteristic that varied in this study was age at the time of intervention, the differences in response could be attributed to the maturity of the individual mice. The micro-CT analysis of the humerus also showed a change in bone volume between adolescent and adult mice, which proves the difference in maturity level. Considering the clinical situation, the typical age of onset of enthesopathy, such as OSD, is at 8–15 years of age (before sexual maturity) [2,3], and the typical age of onset of tendinopathy, such as Jumper’s knee, is at 14–40 years of age (after sexual maturity) [5]. Although there is a slight overlap between the two diseases, it is very likely that factors, such as individual maturity, bone growth, and sexual maturity, contribute to their different ages of onset. If mineralization of the enthesis is insufficient before sexual maturity, the enthesis may be more susceptible to damage than tendons, because the enthesis is a fragile cartilaginous tissue. Our data suggest that with progressive maturation-related mineralization of the fibrocartilage, the enthesis becomes less prone to overuse than the tendon.

This study has some limitations. First, the analysis in this study focused exclusively on inflammatory factors and tissue/morphological changes. It is generally recognized that enthesitis and tendinitis are caused by inflammatory reactions in the enthesis FC region, surrounding tissues, and tendon tissues, which then induce degenerative changes that lead to pathological changes called enthesopathy and tendinosis. However, it has been reported that when degenerative changes occur in the tendon tissue, inflammatory cells are rarely identified at the same time. Interestingly, some reports have suggested that inflammatory responses occur after the degenerative changes [47,48]. In any case, this study only analyzed inflammation-related factors and did not analyze degenerative-related factors; thus, it is not possible to prove that enthesopathy or tendinosis actually occurred or to explain the order, in which they occurred. Furthermore, this study did not test the mechanical strength of the tendons. It is unclear whether tendon hypertrophy is a training effect or an adaptive response to reduced intensity. In addition, different maturity levels should have different exercise tolerances; therefore, running speed variations may be necessary.

However, to the best of our knowledge, few studies have examined the response of tendons and entheses in mice of different ages under different exercise conditions as in this study. In the future, based on the results of this study, we would like to examine whether the effects of inflammatory factors expressed in the entheses and tendons induce degenerative changes, such as enthesopathy and tendinopathy, by considering the running speed as an additional variable, and by conducting long-term interventions and degenerative factor analysis.

## 5. Conclusions

This study suggests that downhill running exercise increases inflammatory factors at the enthesis and tendon, and the site of stress may differ depending on the effect of individual maturity. These results will be useful for elucidating the detailed pathogenesis of sports-related enthesopathy and tendinopathy, which are believed to be caused by common factors with a similar age of onset, and for identifying the factors that contribute to their pathogenesis.

## Figures and Tables

**Figure 1 biology-12-00456-f001:**
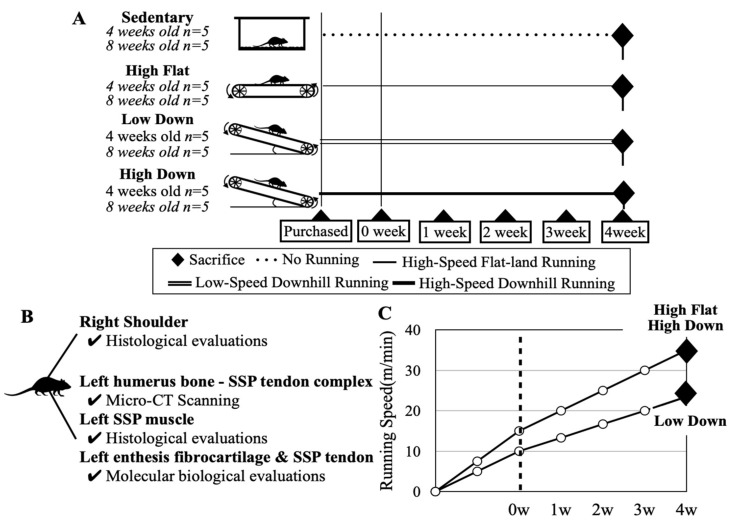
Schematic diagram showing target animals and exercise intervention methods. (**A**) Schematic diagram describing the grouping of target animals and the respective intervention methods and periods. Target mice were purchased at 3 and 7 weeks of age and the intervention was started at 4 and 8 weeks of age, respectively, to compare adolescent and adult mice. All mice were divided into four groups. After 1 week of environmental and exercise adaptation, the mice underwent a 4-week running intervention. Downhill running mimics the eccentric contraction of the supraspinatus muscle. (**B**) Tissues were collected from each mouse. (**C**) The running speed increased in step. The intervention speed did not change for mice of different ages.

**Figure 2 biology-12-00456-f002:**
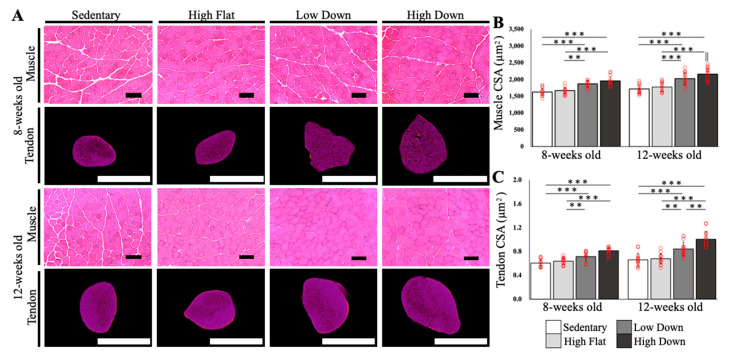
Comparative results of the cross-sectional area of the supraspinatus muscle–tendon complex. (**A**) CSA images of the SSP muscle–tendon according to each week of age are shown. The SSP muscle was imaged by HE staining, and the SSP tendon was imaged by microCT imaging. Scale bar; muscle = 100 µm, tendon = 1 mm. (**B**) Comparison results of the CSA of the SSP muscles. (**C**) Comparison results of the CSA of the SSP tendon. Statistical significance: ** *p* < 0.01, *** *p* < 0.001. || *p* < 0.05 (8-week-old High Down group vs. 12-week-old High Down group). SSP, supraspinatus; HE, hematoxylin and eosin; CT, computed tomography; CSA, cross-sectional area.

**Figure 3 biology-12-00456-f003:**
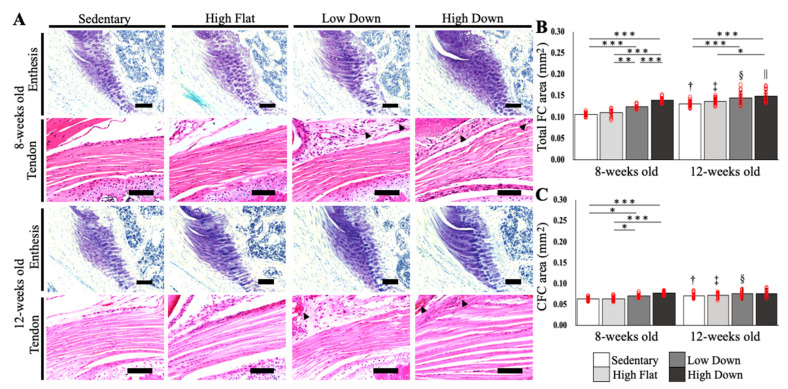
Histological changes in the fibrocartilage of the enthesis and tendon. (**A**) Histological images of TB staining for enthesis FC and HE staining for the tendon are shown. Black arrowhead: angiogenesis. Scale bar; 100 µm. (**B**) Comparison results of the FC area are shown. (**C**) Comparison results of the CFC area. Statistical significance: * *p* < 0.05, ** *p* < 0.01, *** *p* < 0.001. † *p* < 0.05 (8-week-old mice in the Sedentary group vs. 12-week-old mice in the Sedentary group), ‡ *p* < 0.05 (8-week-old mice in the High Flat group vs. 12-week-old mice in the High Flat group), § *p* < 0.05 (8-week-old mice in the Low Down group vs. 12-week-old mice in the Low Down group), || *p* < 0.05 (8-week-old High Down group vs. 12-week-old High Down group). CFC, calcified fibrocartilage; TB, toluidine blue; HE, hematoxylin and eosin.

**Figure 4 biology-12-00456-f004:**
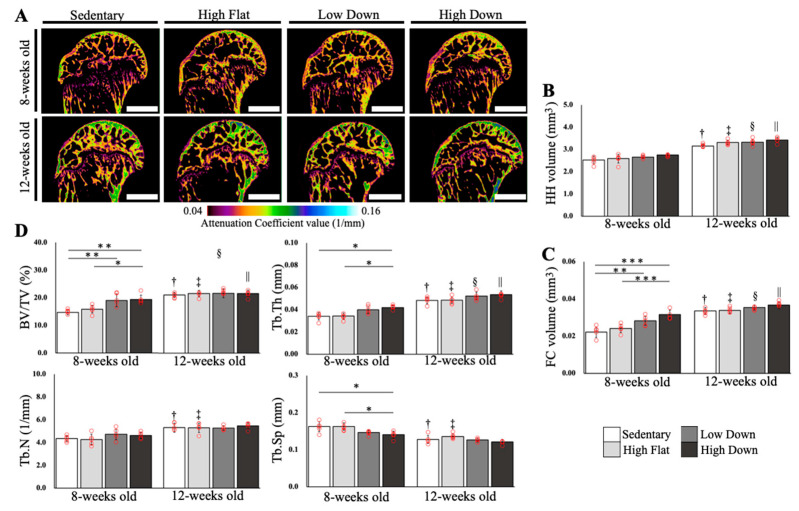
Bone quality evaluation of the humerus head. (**A**) Images of the HH obtained by microCT imaging are shown. Color mapping was performed on the microCT images using the attenuation coefficient (AC) values at the time of imaging as a reference. Scale bar; 1 mm. (**B**) Results of HH volume comparisons. (**C**) Results of the FC volume comparisons. (**D**) Results of bone quality evaluation in the SB region underlying the enthesis CFC area: bone volume/tissue volume (BV/TV), trabecular thickness (Tb.Th), trabecular number (Tb.N), and trabecular separation (Tb.Sp). Statistical significance: * *p* < 0.05, ** *p* < 0.01, *** *p* < 0.001. † *p* < 0.05 (8-week-old mice in the Sedentary group vs. 12-week-old mice in the Sedentary group), ‡ *p* < 0.05 (8-week-old mice in the High Flat group vs. 12-week-old mice in the High Flat group), § *p* < 0.05 (8-week-old mice in the Low Down group vs. 12-week-old mice in the Low Down group), || *p* < 0.05 (8-week-old mice in the High Down group vs. 12-week-old mice in the High Down group). CT, computed tomography; HH, humerus head; CFC, calcified fibrocartilage.

**Figure 5 biology-12-00456-f005:**
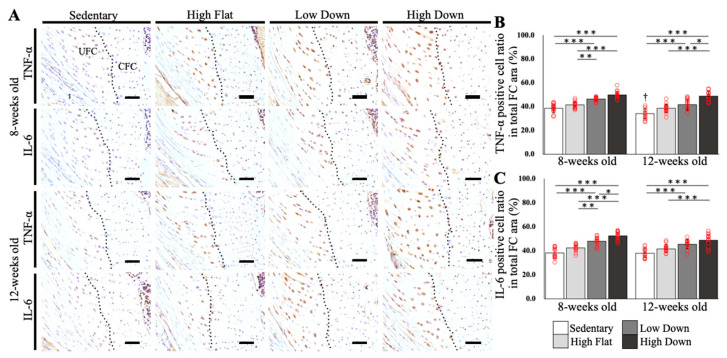
Comparison results of TNF-α and IL-6 expression in the enthesis fibrocartilage (FC) area. (**A**) IHC-stained tissue images of the enthesis FC areas according to week age. TNF-α and IL-6 are used as primary antibodies. The black dot line indicates the tidemark. (**B**) Comparison results of TNF-α expression in the FC area. (**C**) Comparison results of IL-6 expression in the FC area. Statistical significance: * *p* < 0.05, ** *p* < 0.01, *** *p* < 0.001. † *p* < 0.05 (8-week-old mice in the Sedentary group vs. 12-week-old mice in the Sedentary group). TNF, tumor necrosis factor; IL, interleukin; IHC, immunohistochemical; UFC, uncalcified fibrocartilage; CFC, calcified fibrocartilage. Scale bar; 50 µm.

**Figure 6 biology-12-00456-f006:**
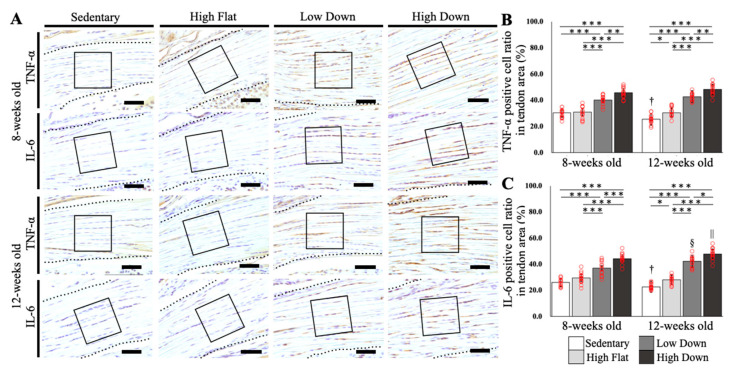
Comparison results of TNF-α and IL-6 expression in the tendon area. (**A**) IHC-stained tissue images of the tendon areas according to week age. TNF-α and IL-6 are used as primary antibodies. The black dot line indicates the tendon. The black box indicates the analysis area. (**B**) Comparison results of TNF-α expression in the tendon area. (**C**) Comparison results of IL-6 expression in the tendon area. Statistical significance: * *p* < 0.05, ** *p* < 0.01, *** *p* < 0.001. † *p* < 0.05 (8-week-old, Sedentary group vs. 12-week-old, Sedentary group), § *p* < 0.05 (8-week-old, Low Down group vs. 12-week-old, Low Down group), || *p* < 0.05 (8-week-old, High Down group vs. 12-week-old, High Down group). TNF, tumor necrosis factor; IL, interleukin; IHC, immunohistochemical. Scale bar: 50 µm.

**Figure 7 biology-12-00456-f007:**
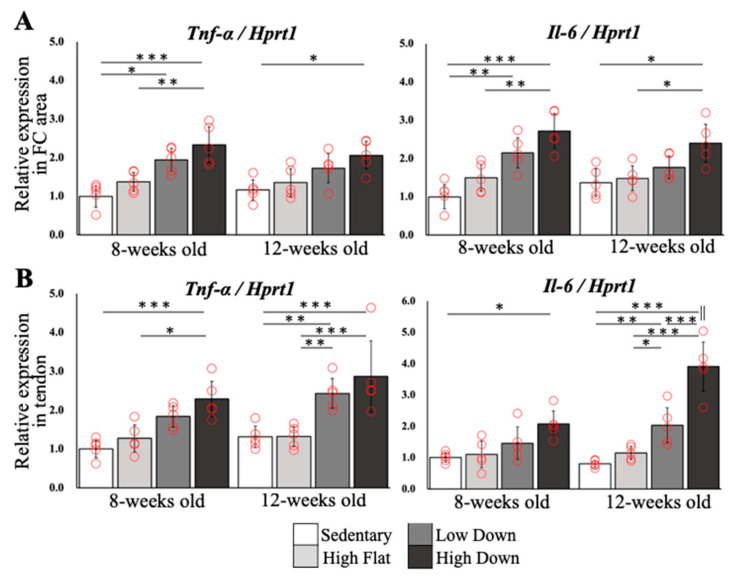
Comparison results of relative gene expression levels. (**A**) Results of comparison of relative gene expression levels of *Tnf-α* and *Il-6* in the enthesis FC area. (**B**) Results of comparison of relative gene expression levels of *Tnf-α* and *Il-6* in tendons. Statistical significance: * *p* < 0.05, ** *p* < 0.01, *** *p* < 0.001. || *p* < 0.05 (8-week-old mice in the High Down group vs. 12-week-old mice in the High Down group).

## Data Availability

The data presented in this study are available in the article and as Appendix A.

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
