# Peer review of "The Effects of Downhill Running and Maturation on Histological and Morphological Properties of Tendon and Enthesis in Mice"

_biology, 2023, doi:10.3390/biology12030456_

Round 1
Reviewer 1 Report
The authors used mouse models subjected to different loading regimens to understand the regulatory factors which drive pathogeneses of tendinopathy and enthesopathy. While the current study provides interesting new information about the effects of muscle contraction type and mouse maturity. However, this study does not provide convincing data to demonstrate obvious tendinopathy and enthesopathy induced by different loading conditions, shown by disorganized collagen fibers, degeneration-related ECM deposition, impaired tissue strength, and so on. The increased FC area, CFC area, and bone quality after both overuse and misuse+overuse might suggest improved function of tendon and enthesis, instead of overuse or misuse caused tissue pathogenesis.
Histological changes in tendon and enthesis at adolescence responding to different loadings, including muscle CSA, CFC area, and expression of inflammatory factors are similar to those at adulthood, especially when there are no statistical comparisons made between adolescence and adulthood. All these results don’t support the conclusion that maturity might mediate tendon and enthesis pathogenesis. Additionally, since the structure, composition, and function of tendon and enthesis at 4 week are different from these of tendon and enthesis at 8 week, evaluation of the effects of tissue maturity on pathogenesis should take the baseline difference into consideration. For example, the result that tnfa gene expression is lower in 8-week mice after misuse+overuse than 12-week mice after misuse+overuse doesn’t demonstrate more pathogenesis in 8-week mice after muscle contraction, because tnfa gene expression is lower in 4-week mice than 8-week mice. Therefore, more and better statistical analyses should help address this weakness.
Specific comments
1. The text size is too small to read in Fig. 1.
2. It is interesting to find that overuse and misuse+overuse increase tendon CSA, FC area, MFC area, and bone quality (suggesting better tissue function responding to loading), but more expression of inflammatory factors (suggesting tissue regeneration). Additional discussion about these conflicting results would be helpful.
Author Response
Dear Reviewer 1
Thank you for reviewing the manuscript and for providing your valuable comments. We have responded to your questions and comments in a point-by-point manner below. Black text indicates a comment from you and red text is my response. The red text in the manuscript shows the text that I modified based on the reviewers' comments, and the blue text shows the text that I changed for similarity.
Comments and Suggestions for Authors
The authors used mouse models subjected to different loading regimens to understand the regulatory factors which drive pathogeneses of tendinopathy and enthesopathy. While the current study provides interesting new information about the effects of muscle contraction type and mouse maturity. However, this study does not provide convincing data to demonstrate obvious tendinopathy and enthesopathy induced by different loading conditions, shown by disorganized collagen fibers, degeneration-related ECM deposition, impaired tissue strength, and so on. The increased FC area, CFC area, and bone quality after both overuse and misuse+overuse might suggest improved function of tendon and enthesis, instead of overuse or misuse caused tissue pathogenesis.
- Response:
-  As you have pointed out, the results of this study did not allow us to verify the factors related to degeneration, disruption of collagen fibers by picrosirius red staining, or confirmation of tissue strength by mechanical testing. Therefore, we can only report that inflammatory reactions occurred in the tendon and in the enthesis, and morphological changes occurred in the enthesis. Therefore, the present study did not actually prove that pathological changes had occurred as in enthesopathy. We have restructured the manuscript to focus on the evaluation of changes occurring relative to enthesitis and tendinitis only. Nonetheless, it is common in enthesopathy for inflammation to occur simultaneously with morphological changes in and around the parenchyma of the enthesis. When mechanical stress occurs in the same area, an increase in inflammatory factors such as TNFα and IL-17 is first observed. Subsequently, cartilage synthesis and bone formation are enhanced in the same area1–3. Bone formation has been reported to be accelerated in correlation with the inflammatory response in the same area4. In addition, considering the relatively young age of animals used in this study, it would be expected that this simultaneous response is more likely. Although we did not identify any degenerative or calcification-related factors in this study, we cannot prove that degenerative changes occurred. We have reported previously that there was increased MM P-13 expression, increased Collagen type X, and a predominant increase in alkaline phosphatase in the same area5. This is in line with the results of a similar study by Wang et al. in which enthesopathy-like morphological changes were observed with the application of abnormal mechanical stress6. Therefore, we are convinced that the results of this study are consistent with those reported previously regarding the occurrence of inflammatory reactions in enthesis and the morphological changes caused by mechanical stress on the same area and that the data provide convincing evidence that enthesitis occurred.
Histological changes in tendon and enthesis at adolescence responding to different loadings, including muscle CSA, CFC area, and expression of inflammatory factors are similar to those at adulthood, especially when there are no statistical comparisons made between adolescence and adulthood. All these results don’t support the conclusion that maturity might mediate tendon and enthesis pathogenesis. Additionally, since the structure, composition, and function of tendon and enthesis at 4 week are different from these of tendon and enthesis at 8 week, evaluation of the effects of tissue maturity on pathogenesis should take the baseline difference into consideration. For example, the result that tnfa gene expression is lower in 8-week mice after misuse+overuse than 12-week mice after misuse+overuse doesn’t demonstrate more pathogenesis in 8-week mice after muscle contraction, because tnfa gene expression is lower in 4-week mice than 8-week mice. Therefore, more and better statistical analyses should help address this weakness.
- Response:
-  Thank you for your suggestions. In the initial draft, all intervention groups were analyzed using one-way ANOVA, and in the revised manuscript, the analyses have been conducted using two-way ANOVA. We also clarified the difference between adolescence and adulthood in testing for main effects and interactions to determine whether the changes were caused by the age in weeks or the intervention method. Furthermore, for the post hoc tests, we used Bonferroni methods and performed multiple comparisons.
-  There is a difference in the baseline as you have pointed out, and the baseline morphology may have been different in the control group because the effect of age in weeks was more pronounced in the control group with regard to the FC area and bone volume. However, there were no significant differences in muscle CSA, tendon CSA, or PCR expression of TNF-a and Il-6 between the 8- and 12-week control groups. Therefore, these findings emphasize that there were no baseline differences in muscle-tendon morphology or inflammation that may have influenced the results of this study. In addition, the 8-week control group was set as the baseline value for PCR, and the 2-ΔΔCT values of the other groups were compared to this baseline value. Therefore, if a significant increase was found in the Misuse+Overuse group at 12 weeks compared to that in the Misuse+Overuse group at 8 weeks, it simply suggests that the inflammation worsened in Misuse+Overuse groups at 12 weeks owing to its higher expression.
- Revised Lines; 188-195 “Statistical analysis was performed using JASP 0.16.3 (Intel)[22,23]. Normality of distribution in each dataset was confirmed using the Shapiro-Wilk test. All data were subjected to a two-way analysis of variance to determine significant effects for each pa-rameter. The main effects and interactions for age in weeks (8- and 12- weeks old) and intervention method (Control, Overuse, Misuse, Misuse+Overuse). Simple main-effect tests were performed on the results for which an interaction was confirmed. The Bon-ferroni test was used as a post hoc test. All data are presented as mean ± standard deviation (SD), with p < 0.05 being statistically significant.”
Specific comments
1. The text size is too small to read in Fig. 1.
- Response:
-  Thank you for pointing this out. We have enlarged the text size and changed the position of panels A-C so that the entire figure is larger.
- Revised Lines; 112
2. It is interesting to find that overuse and misuse+overuse increase tendon CSA, FC area, MFC area, and bone quality (suggesting better tissue function responding to loading), but more expression of inflammatory factors (suggesting tissue regeneration). Additional discussion about these conflicting results would be helpful.
- Response:
-  Thank you for your question. Our response is the same as that for the first comment. Previous studies have shown that morphological changes in the enthesis and promotion of inflammatory responses in the same region have been reported in many studies related to enthesitis, such as spondyloarthritis. It is also known that bone and cartilage formation in the enthesis is accelerated in correlation with the degree of inflammatory response that occurs in the enthesis; therefore, we are confident that the changes that occurred in this study are consistent with these previous findings. An explanations of these details has been added to the Discussion, along with citations of previous studies.
- Revised Lines; 412-424
-  “The link between inflammation development and morphological changes in the enthesis has been reported in studies related to psoriatic arthritis and spondylarthritis, which are representative diseases of enthesitis. Triggered by mechanical stress or infection, the enthesis induces IL-23 release, which in turn activates ILC3 and γδ T cells[33,34]. Subsequently, γδT cells and ILC3 increase the expression of inflammatory factors such as TNF-α and IL-17, IL-22, which induce the activation of mesenchymal stem cells and osteogenesis-related signaling activated in a mechanical stress-dependent manner, and leads to morphological changes in the tissues[35,36]. Although we were unable to analyze IL-23 levels in our study, we observed increased TNF-α expression and morphological changes in the enthesis, and in our previous report, we confirmed activation of osteogenesis-related signaling pathways (BMPs) in the same region[5]. Therefore, it is possible that an inflammatory response triggered by increased mechanical stress specific for EC was induced in the enthesis, leading to enthesopathy-like morphological changes.”

Reviewer 2 Report
The present work is an original article within the scope of this journal. It is, as also stated by the authors, based on previous research performed by the group, as often stated in the manuscript and is a significant early step that could contribute in managing certain health conditions. It has used established methods and clearly provide the results of the experimental work. In general, it is an interesting research work, easy to read and comprehended by the readers, as whole.
In particular though, there some points that seem to need more clarification and/or rephrasing.
1). As a general remark, this is an original paper, relying as stated often by the authors, much on previous work performed by the group. It is important though, for the reader, to be able to easily follow the text without feeling the need to be familiar with and/or retract to the group’s previous work.
Under this note, rephrasing, adding brief explanatory comments and if possible, providing additional references, independent of the group’s previous work, should be considered by the authors in particular in the sections stated below:
Lines 59-62 …“been demonstrated that the type of muscle contraction during movement, rather than the amount of exercise, is a more important factor in enthesopathy pathogenesis. Additionally, the activation of inflammatory factors in tendons was also confirmed…..”
The use of certain affirmative words (demonstrated, confirmed) in a section that mainly refers to the group’s previous recent experimental work, one cannot be sure, when not familiar with this previous work, how this was established when there is no other independent research reference(s) to support this finding. Therefore, I would suggest, to either rephrase this text or further support the statement eg providing further bibliography and/or explanation.
Line 65 -This reflects as a weak statement since it is supported by only one reference. As this statement is part of the initial hypothesis of the study it should be further supported. Please add some more supportive information and (if available), references.
2) Materials and Results section
Lines 94-96 refer to the exercise protocol used for the mice. It is not clear how this was established and if the whole protocol was based on the previous work as refered or only part of it. Please clarify this section, as to how the experimental parameters were established, in particular duration and type of exercise (use of previous established protocols? Optimization experiments?).
3) Simple Summary and Conclusion
Lines 22-24 and lines 433-435. In both of these occasions, the same claim is made as it is suggested by the authors that “In clinical practice, athletes who exhibit enthesopathy or tendinopathy they should first be evaluated for predominance of muscle contraction type and then age…” according to their current research findings. The authors should maybe consider rephrasing this section as the data of one research team performed in mice are difficult to be extrapolated in humans. It seems as a premature statement and the data available as the authors state (lines 428-29) are from a study that is the basis of further research. So, it feels like this is an overstatement and should be rephrased or eliminated.
Non Published material:
Supplementary Methods:
Line 10. “Care was taken to ensure bone was not include”. Please provide brief information of how this was ensured as bone marrow contamination may effect gene expression measurements
Author Response
Dear Reviewer 2
Thank you for reviewing the manuscript and for providing your valuable comments. We have responded to your questions and comments in a point-by-point manner below. Black text indicates a comment from you and red text is my response. The red text in the manuscript shows the text that I modified based on the reviewers' comments, and the blue text shows the text that I changed for similarity.
Comments and Suggestions for Authors
1) As a general remark, this is an original paper, relying as stated often by the authors, much on previous work performed by the group. It is important though, for the reader, to be able to easily follow the text without feeling the need to be familiar with and/or retract to the group’s previous work.
 Under this note, rephrasing, adding brief explanatory comments and if possible, providing additional references, independent of the group’s previous work, should be considered by the authors in particular in the sections stated below:
 Lines 59-62 …“been demonstrated that the type of muscle contraction during movement, rather than the amount of exercise, is a more important factor in enthesopathy pathogenesis. Additionally, the activation of inflammatory factors in tendons was also confirmed…..”
 The use of certain affirmative words (demonstrated, confirmed) in a section that mainly refers to the group’s previous recent experimental work, one cannot be sure, when not familiar with this previous work, how this was established when there is no other independent research reference(s) to support this finding. Therefore, I would suggest, to either rephrase this text or further support the statement eg providing further bibliography and/or explanation.
- Response:
-  Thank you for your valuable input. Accordingly, we have paraphrased the text and added the following references to support our study in lines 59-62.
- Revised Lines; 52-65
-  “Previous studies have suggested that sports-related pathological changes similar to enthesopathy are more strongly influenced by significant eccentric-contraction (EC)-dominant movements associated with misuse of the body than by simple increases in physical activity, as suggested by basic experiments in young mice, regardless of the magnitude of the exercise load. EC-dominant movements induce increases in inflammatory factors in the enthesis and tendons, and significant increases in the cartilage catabolic factor (matrix metalloproteinase (MMP)-13), cartilage synthesis-related factor (SRY-box9; Sox9), and calcification-related factors (collagen type X, ColX; alkaline phosphatase, ALP; runt-related transcription factor, Runx2) in the enthesis [4][5]. In fact, EC has been reported to be a specific contraction type that when compared to other types of muscle contraction (concentric contraction, CC; isometric contraction, IC), is more likely to cause mechanical stress on muscle and tendon tissue, and to increase inflammatory factors [6][7]. Therefore, it is possible that changes in muscle contraction type dominance during movement may have effects on the enthesis and the tendon substance”
2) Materials and Results section
 Lines 94-96 refer to the exercise protocol used for the mice. It is not clear how this was established and if the whole protocol was based on the previous work as refered or only part of it. Please clarify this section, as to how the experimental parameters were established, in particular duration and type of exercise (use of previous established protocols? Optimization experiments?).
- Response:
-  The exercise protocol was established based on our previous results and studies, as well as preliminary experiments. Regarding the most important type of exercise, downhill running commonly induces eccentric contraction of antigravity muscles. This experimental system is also described in the “Resource Book for the Design of Animal Exercise Protocols (2006)”1. In addition, the exercise format for inducing eccentric contraction has been used in many previous studies2, such as that by Soslowsky et al3. based on the study by Armstrong et al4. Next, regarding the exercise duration, our intention in this study was to mimic high-intensity exercise. In exercise experiments using mice and rats, the most commonly used moderate-intensity exercise was set at a speed of 12-18 m/min and an exercise duration of 30 to 50 min5–8. In this study, the maximum exercise intensity for the Overuse and Misuse+Overuse groups was 35/min and 1 h, respectively, which was clearly at a higher intensity than the interventions performed in other studies. Furthermore, in our preliminary experiments, we confirmed that many 4-week-old mice dropped out when running at 20 m/min for 1 hour and more than half of the ten 8-week-old mice dropped out when running at 40 m/min for 1 hour, because they could not keep up with the speed. Therefore, we set the maximum intervention speed during the intervention period as 35 m/min and the starting speed as 15 m/min and established our own protocol to gradually increase the speed as the mice grew. It is clear that the exercise speed in this study was faster and more excessive than that used in typical high-intensity exercise groups (25 m/min). We have added the above information to the revised manuscript.
- Revised Lines; 95-102
-  “The upper limit of the intervention speed was set at 35 m/min at the end of the intervention because many 8- and 12-week-old mice dropped out at speeds faster than 35 m/min during the 1-hour dropout test. This intervention speed was faster than the speed generally used for high-intensity exercise (25 m/min)[16]. The intervention start-speed was established at 15 m/min as a moderate speed because many mice dropped out at 20 m/min during the 1-hour dropout test at age 4-week. Therefore, the starting speed of the intervention was set at 15 m/min and the speed at the end of the intervention at 35 m/min, and the exercise conditions were set to gradually increase with growth.”
3) Simple Summary and Conclusion
 Lines 22-24 and lines 433-435. In both of these occasions, the same claim is made as it is suggested by the authors that “In clinical practice, athletes who exhibit enthesopathy or tendinopathy they should first be evaluated for predominance of muscle contraction type and then age…” according to their current research findings. The authors should maybe consider rephrasing this section as the data of one research team performed in mice are difficult to be extrapolated in humans. It seems as a premature statement and the data available as the authors state (lines 428-29) are from a study that is the basis of further research. So, it feels like this is an overstatement and should be rephrased or eliminated.
- Response:
-  We have adjusted our claims accordingly to avoid any overstatement. We fully understand that the results of this study alone cannot be generalized to humans. Thus, we have removed the indicated sentence and have taken care to describe only that which can be effectively conveyed by this study alone.
Supplementary Methods:
 Line 10. “Care was taken to ensure bone was not include”. Please provide brief information of how this was ensured as bone marrow contamination may effect gene expression measurements.
- Response:
-  Thank you for pointing this out. It is our understanding as well that bone marrow contamination can have a considerable impact on gene expression. Therefore, the collection was performed with a microsurgical knife under a stereomicroscope. The bone marrow was always red because the sample was collected in as fresh a state as possible. We ensured that the red tissue was not contaminated through macroscopic observation under a stereomicroscope with the maximum possible magnification of the extracted tissue. The following text was added.
- Revised Lines; Supplemental file 10-11
-  “The samples were carefully collected with observation under a stereomicroscope to ensure that no bone marrow was included.”

Round 2
Reviewer 1 Report
The authors have well addressed my comments and the manuscript has been significantly improved.
Author Response
Reviewer: 1
Review’s Comments for Author(s)
The authors have well addressed my comments and the manuscript has been significantly improved.
Response: Thank you for taking the time to perform a meticulous review of our paper. Your valuable comments and advice were very helpful for preparing this paper. Thank you very much.